# Application of Bionic Technologies on the Fracturing Plug

**DOI:** 10.3390/biomimetics4040078

**Published:** 2019-12-05

**Authors:** Lin Chen, Ran Wei, Songbo Wei, Xinzhong Wang

**Affiliations:** PetroChina Research Institute of Petroleum Exploration and Development, Beijing 100083, China; chenlin_zb@petrochina.com.cn (L.C.); weiran2012@petrochina.com.cn (R.W.); wxzh@petrochina.com.cn (X.W.)

**Keywords:** fracturing, dissolvable bridge plug, surface dimple, dissolution rate

## Abstract

The dissolvable bridge plug is one of the most important tools for multi-stage hydraulic fracturing in the field of oil/gas development. The plug provides zonal isolation to realize staged stimulation and, after fracturing, the plug is fully dissolved in produced liquids. A bionic surface was introduced to improve the performance of the plug. Surface dimples in the micron dimension were prepared on the dissolvable materials of the plug. The experimental results showed that the surface dimples changed the hydrophilic and hydrophobic properties of the dissolvable materials. The dissolution rate has a great relation with the parameters of the dimples and can be controlled by choosing the dimples’ parameters to some degree.

## 1. Introduction

The rapid development of unconventional oil/gas has been led to tremendous changes in the global energy industry. According to the U.S. Energy Information Administration’s International Energy Outlook 2016, the production of unconventional gases, including tight gas, shale gas, and coal bed methane, will increase from 0.8227 trillion cubic meters in 2015 to 2.48 trillion cubic meters in 2040. Over the same period, the production of unconventional oil, including tight oil, shale oil, extra-heavy oil, field condensate, and bitumen, will increase from 0.480 billion tons to 1 billion tons [1]. As one of the key technologies used in unconventional oil/gas development, multi-stage hydraulic fracturing technology of horizontal wells can effectively increase production.

Hydraulic pumping bridge plug staged fracturing technology has the advantages of a reliable seal, unlimited staged fracturing, and a short fracturing cycle. During the fracturing operations, the plug is pumped down and set on the first target position, and then the perforating and fracturing processes are carried out successively. The same fracturing operation is repeated for the remaining stages. Conventional drillable plugs are widely used today, but the drilling of conventional plugs is time-consuming, expensive, and high risk [1].

Currently, the dissolvable bridge plug has been developed, and this kind of plug can dissolve in wellbore fluid, which eliminates the plug-drilling procedure [2,3]. Once the dissolvable plugs contact the well liquids in the wellbore, the dissolving process will begin. Generally, the early pump-down plugs are severely corroded, especially for horizontal wells with lots of stages. In this situation, the high released pressure of the fracture zone will cause many unpredicted difficulties and risks to the ongoing operations. Therefore, the early pump-down plugs should not dissolve too quickly, and the dissolving rates of all plugs in the wellbore should be controlled to some degree.

There are many methods to affect the dissolving rate of the plug, such as the material composition, surface coating, structure design, etc. Surface treatment is one of the most important methods to influence the dissolving rate of the material [4].

Nature is a source of innovations and solutions for many technological problems, and it has inspired the development of technologies in many fields. In recent years, in the material surfaces field, bio-inspired functional surfaces have been a very hot research topic [5,6,7,8,9]. It is well known that the lotus leaf has a natural hydrophobic behavior due to the surface topography of the lotus leaves having a special roughness structure, which is strongly related to the hydrophobicity of the lotus leaf. Inspired by that, designing microstructures of dissolvable material surfaces might be an effective way to change the contact status with the surrounding fluid and thus influencing the initial dissolving rate of the plug.

In this paper, the dimple surface textures were fabricated on the dissolvable magnesium alloy substrates by a laser surface texturing technique. The surface morphologies of the symmetrically distributed dimple surface textures were observed by an optical microscope and white-light interferometer. The microstructure of the dimple unit and the chemical compositions were characterized by a scanning electron microscope and energy-dispersive X-ray spectrometer. The water contact angles and the dissolution characteristics of the samples with and without dimple surface textures were then tested. The relationship between the wettability and the dissolution rate was discussed.

## 2. Materials and Methods

### 2.1. Fabrication of Dimple Surface Textures (DST)

Dissolvable magnesium alloy samples with dimensions of Φ45 mm × 4 mm were used as the substrate materials. For the material specifications, refer to Pei’s report [10]. Laser beam machining was used to fabricate the symmetrically distributed DST. The laser source used was a fiber nanosecond pulse laser with a maximum power output of 20 W and a center wavelength (λ) of 1064 nm. The dimple unit diameter of Φ100 μm was designed, and the area densities of the dimples were 10%, 20%, and 30%, respectively. The laser machining process was conducted in the atmosphere.

### 2.2. Characterizations of Surface Morphology and Wettability

The surface morphologies of the DST samples were examined by using an optical microscope. The three-dimensional micrograph of the dimple surface texture was characterized by using a white-light interferometer. The morphology and size of the samples were observed by JSM-7001F scanning electron microscopy at 15 K eV. The chemical compositions were analyzed by using an energy-dispersive X-ray spectrometer. The water contact angles were measured by using a contact angle meter with a 5 μL water drop at room temperature. The average values of the water contact angles were obtained from at least three measurement positions on each sample.

### 2.3. Dissolution Tests

The samples with and without dimple surface textures were immersed in the 2% KCl solutions at temperatures of 85 °C. All samples were taken out from the solutions every 6 h and were dried by using a hot-air blower. Then, the remaining weight of the samples was measured by using a precise electronic balance.

## 3. Results and Discussion

Figure 1 shows the optical micrographs of the dimple textures on dissolvable materials with areal densities of 10%, 20%, and 30%. It can be seen that all dimple surface textures were consistent in appearance and were uniformly distributed on the sample surface. With an increase in the areal density of the dimple surface texture, the distance between the neighboring dimples decreased. Figure 2 shows the cross-section profile and the three-dimensional micrograph of the dimple surface texture with an areal density of 10%. The three dimensional space structure of the dimple unit was like an inverted cone. The internal surfaces of the dimples were relatively smooth. The maximum depth of the dimple unit was about 110 μm.

Figure 3a shows the SEM image of the dimple surface texture with an areal density of 10%. Figure 3b shows the EDS analysis of the laser melting region around the dimple unit. Four elements—Mg, Al, O, and C—were detected by EDS, and the oxygen content reached 48 at%, which meant that the MgO compositions were produced during laser processing.

Figure 4 shows the water contact angle on the different samples of dissolvable materials. Figure 4a shows the water contact angle on the surface without DST, where the contact angle was about 84°. Figure 4b shows the contact angle for the sample with a DST areal density of 10%, where the contact angle was about 138°. Figure 4c shows the contact angle for the DST sample with an areal density of 20%, where the contact angle was about 143°. Figure 4d shows the contact angle for the DST sample with an areal density of 30%, where the contact angle was about 150°. It can be seen that the surface without DST on dissolvable materials was hydrophilic, while the surface with a dimple surface texture exhibited hydrophobic properties.

Explanations for the high hydrophobicity of the surface have been proposed by several theories and models. The Cassie–Baxter model is a simplified way to explain how the rough surface structures achieve good hydrophobic properties. When a water droplet contacts nano- and micro-scale surface structures, the droplet only contacts some parts of the surface due to the special features, leaving air trapped between the droplet–surface interfaces [11,12,13,14,15]. In the present work, the hydrophobicity of dissolvable material surfaces with DST can be understood from the above theory. Under the droplet, the air was trapped in the dimple, and the droplet was supported by parts of the material surface and the trapped air in the dimple. In addition to the surface microstructures, the laser process brought changes to the chemical composition on the surface, which also affected the wettability of the surface to some extent [16]. 

Figure 5 shows the surface morphologies after 1.5 h of immersion in KCl solution for different samples. The white parts covered with MgO compositions on the surface of the samples represent the corroded areas. It can be seen from Figure 5a that for the sample without DST, most of the sample experienced the corrosion phenomenon. The corrosion degree for the samples with DST was less than that of the sample without DST, and the corrosion only happened on a small area.

Figure 6 shows the surface morphologies after 12 h of immersion in KCl solution for different samples. All samples underwent total corrosion, and the dimples disappeared for the DST samples. It is difficult to observe the differences among the four samples.

Figure 7 shows the weights of different samples as a function of the immersion time in KCl solution. In the initial 6 h, the sample without DST lost about 1.2 g, the sample with a DST areal density of 10% lost about 1.0 g, and the samples with DST areal densities of 20% and 30% lost about 0.7 g. The dissolution rates of the samples with DST areal densities of 20% and 30% were about 58.3% that of the sample without DST in the initial dissolving stage.

Along with an increasing immersion time, the weight of the samples without DST decreased faster than that of other samples. Additionally, the weight loss of the samples with a DST areal density of 10% was less than that of samples without DST but was more than that of samples with DST areal densities of 20% and 30%. It seems that the weight change curve for the samples with DST areal densities of 20% and 30% were almost coincident, which means that areal densities of both 20% and 30% had no significant differences in dissolution rates.

After 24 h, the sample without DST lost about 4.6 g, the sample with a DST areal density of 10% lost about 3.9 g, and the samples with DST areal densities of 20% and 30% lost about 3.1 g. For the whole 24 h, the average dissolution rates of the samples with DST areal densities of 10% and 30% were about 84.8% and 67.4% that of the sample without DST, respectively. For the last 12 h, the average dissolution rates of the samples with DST areal densities of 10% and 30% were about 94.7% and 73.7% that of the sample without DST, respectively. This suggests that when the DST layer was destroyed by dissolution, the dissolution rates of the samples with DST began to approach that of the sample without DST.

The fact that DST influenced the dissolution rate can be understood from two aspects. Firstly, the DST samples have hydrophobic properties, which led to less real area of the substrate directly contacting the liquids. In particular, with an increase in the DST areal density, the real contact area between liquids and dissolvable substrate decreased correspondingly, thus less material surface participated in the dissolving reaction. Secondly, in the laser melting area, MgO compositions were produced during the laser process, so less metal magnesium was exposed to the liquids to participate in chemical reactions. Due to the above two reasons, the DST significantly reduced the dissolution rate in the initial immersion stage. Along with increasing the immersion time, the metal magnesium area constantly corroded, and finally, the DST layer was destroyed and disappeared thoroughly. The dissolution rate of DST samples reached that of normal magnesium metal.

## 4. Conclusions

In this study, samples with dimple surface textures were fabricated on the dissolvable magnesium alloy substrate by using a laser surface texturing technique. The surface morphologies and the wettability of the DST samples with different areal densities were observed and measured. The experimental results showed that the DST could increase the water contact angle. The water contact angle on the surface without DST was about 84°. The water contact angle increased from 138° to 150° when the DST areal density increased from 10% to 30%. In the initial immersion stage, the DST significantly reduced the dissolution rate, and the dissolution rates of the samples with DST areal densities of 20% and 30% were about 58.3% that of the sample without DST. When the DST layer was destroyed by dissolution, the dissolution rates of the samples with DST began to approach that of the sample without DST. The areal densities of both 20% and 30% had no significant effect on the dissolution rates. The hydrophobic properties of the DST surface and the change in the chemical composition were the main reasons for the decrease in the dissolution rate of the dissolvable materials.

## Figures and Tables

**Figure 1 biomimetics-04-00078-f001:**
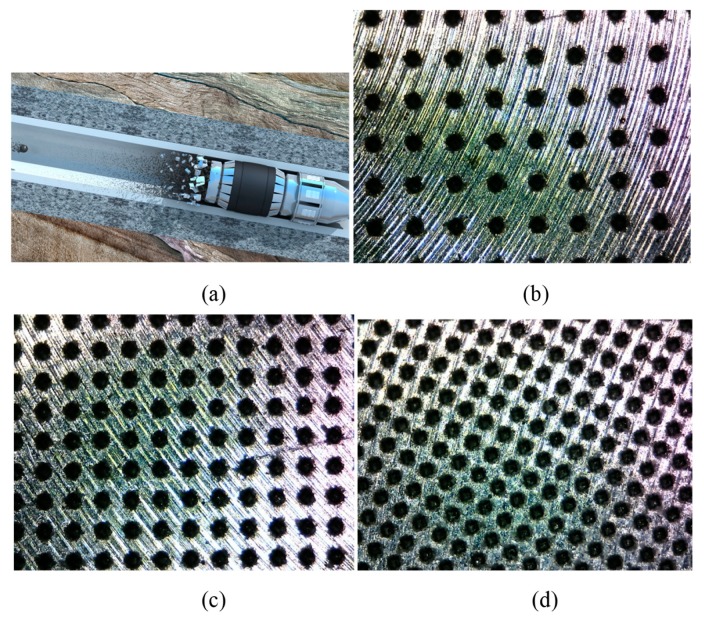
Dissolving schematic diagram of the dissolvable bridge plug (**a**). The optical micrographs of dimple surface textures on the dissolvable materials with areal densities of (**b**) 10%, (**c**) 20%, and (**d**) 30%.

**Figure 2 biomimetics-04-00078-f002:**
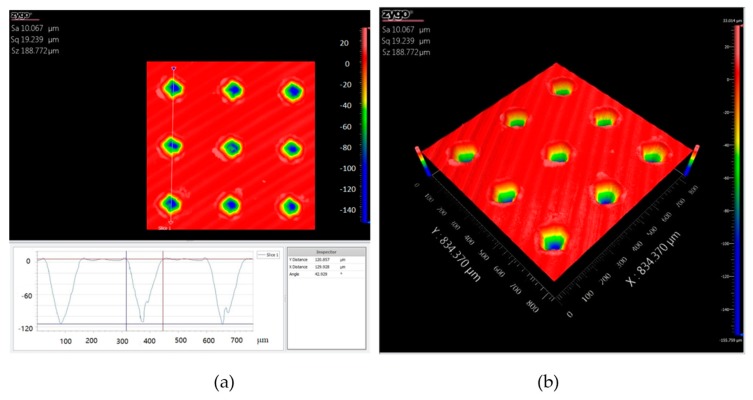
The cross-section profile (**a**) and three-dimensional micrograph (**b**) of the dimple surface textures with an areal density of 10%.

**Figure 3 biomimetics-04-00078-f003:**
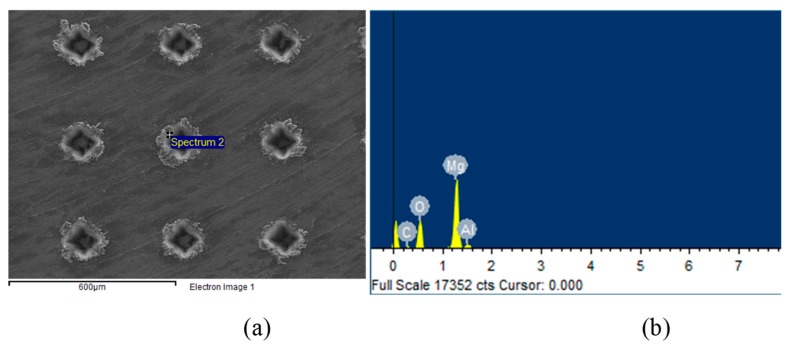
(**a**) SEM image of the dimple surface texture with an areal density of 10%, (**b**) EDS analysis of the laser melting region around the dimple unit.

**Figure 4 biomimetics-04-00078-f004:**
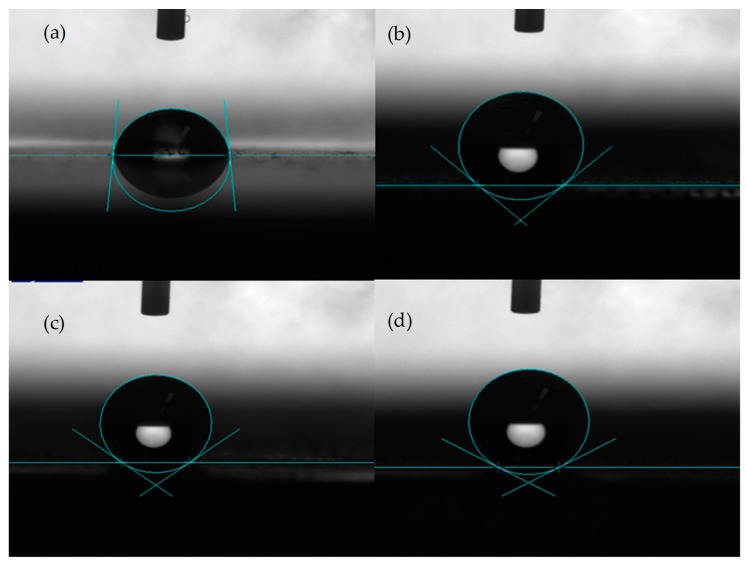
Water contact angle on the sample without dimple surface texture (DST) (**a**), with DST areal densities of (**b**) 10%, (**c**) 20%, and (**d**) 30%.

**Figure 5 biomimetics-04-00078-f005:**
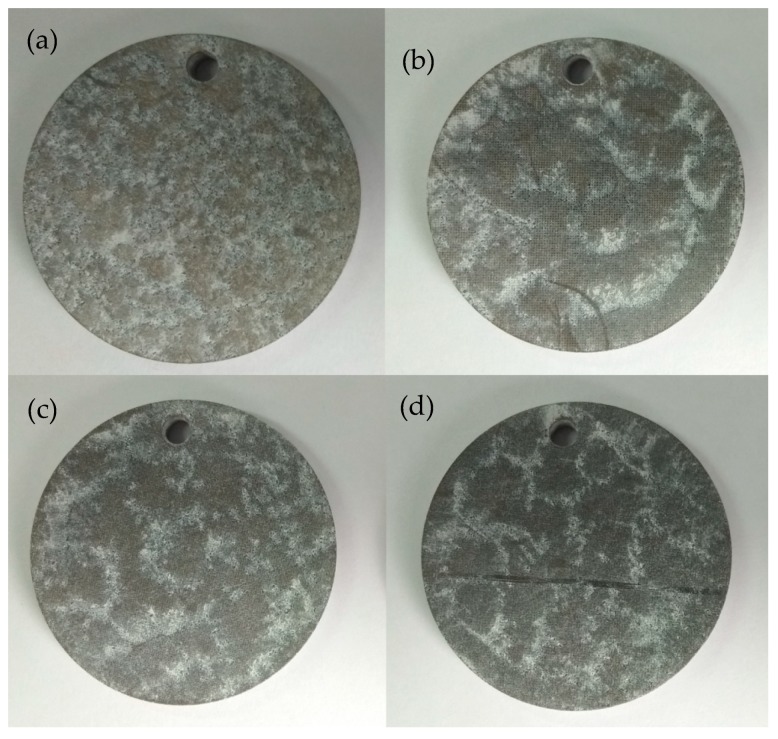
Surface morphologies after 1.5 h of immersion in KCl solution for different samples: (**a**) without DST, with DST areal densities of (**b**) 10%, (**c**) 20%, and (**d**) 30%.

**Figure 6 biomimetics-04-00078-f006:**
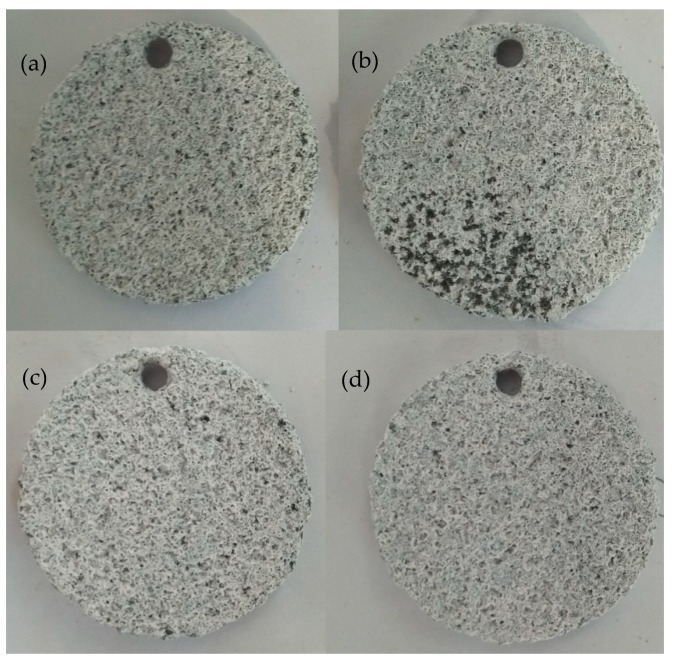
Surface morphologies after 12 h of immersion in KCl solution for different samples: (**a**) without DST, with DST areal densities of (**b**)10%, (**c**) 20%, and (**d**) 30%.

**Figure 7 biomimetics-04-00078-f007:**
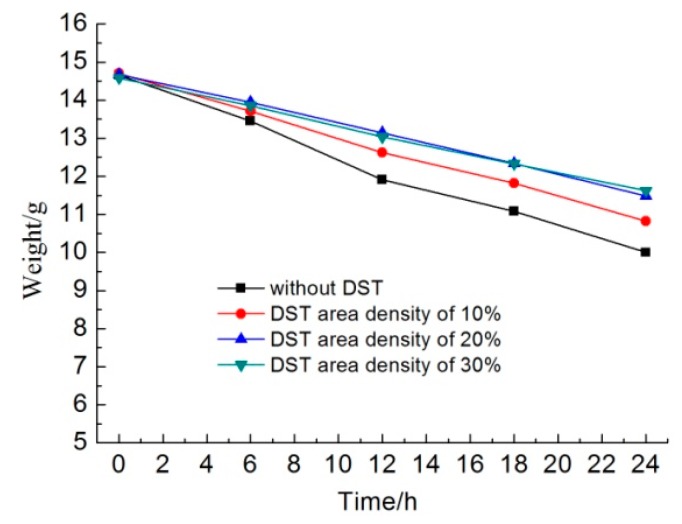
The weights of different samples as a function of the immersion time in KCl solution.

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
