# Peer review of "Application of Bionic Technologies on the Fracturing Plug"

_biomimetics, 2019, doi:10.3390/biomimetics4040078_

Round 1

Reviewer 1 Report

The authors present a method to modify the wetability of a dissolvable bridge plug by patterning the surface with dimples. The method is inspired by the lotus effect. The overall quality of the paper is mediocre. There are multiple aspects that require improvement.

In general, please improve the grammar and style of writing.

1. Introduction:

The introduction clearly states the background and importance of the work. Though, it is not quite clear what a bridge plug is and how it looks. Please add a detailed image of a typical bridge plug and mark the Magnesium part that you did the modifications to.

2. Materials and Methods

All relevant aspects are mentioned. However, the description of the single methods and materials lacks some detailed information. Please add

Specification of magnesium alloy. Precise description of the laser machining process including process parameters. Has inert gas been used for laser machining? Why did you use 2% KCl solution at 85°C for dissolution?

3. Results and Discussion

Figure 2 must be improved. The y-axis on the profilogram as well as the color coding for the 3D-images are barely readable.

The EDS analysis is clear. However, I expect a comparison with the non-machined area of the surface to prove that the formation of MgO was caused by laser machining.

Figure 3 (b) must be improved. The spectrum does not provide sufficient information. Please put a table of atomic percentage of elements instead.

The measurement and description of contact angle measurement is clear and does not lack any information. The interpretation of the results is appropriate.

The results of the dissolution experiment are clear and precisely described. In figure 5, please add SEM images or 3D micrographs to the corresponding images, if available. Photographs do not allow precise analysis of the morphology other than a rough estimation of coroded area.

The discussion of the results is clear and appropriate.

4. Conclusion

The major findings of the work are well summarized. However, there is no conclusion or discussion of the results.

How do the results compare to similar approaches, if available? How do the results compare to other methods for the modification of dissolvability of bridge plugs that you mentioned (surface coating, change of material composition, etc.)? Can the presented method be applied to real bridge plugs? Do you expect any changes or improvement in performance when adjusting the DST parameters (e.g. diameter, depth)?

Author Response

Thanks very much for your review and good suggestions.

Introduction:

The introduction clearly states the background and importance of the work. Though, it is not quite clear what a bridge plug is and how it looks. Please add a detailed image of a typical bridge plug and mark the Magnesium part that you did the modifications to.

Reply: Figure 1(a) shows the bridge plug, the dimple surface textures will be designed on the outside surface of all magnesium parts.

Materials and Methods

All relevant aspects are mentioned. However, the description of the single methods and materials lacks some detailed information. Please add Specification of magnesium alloy. Precise description of the laser machining process including process parameters. Has inert gas been used for laser machining? Why did you use 2% KCl solution at 85°C for dissolution?

Reply: In the revised manuscript we add some specifications of magnesium alloy and the laser machining process. No inert gas was used for laser machining. The mineralization degree and temperature of formation water are diverse for different areas, in this work we just choose a common condition to conduct the tests.

Results and Discussion

Figure 2 must be improved. The y-axis on the profilogram as well as the color coding for the 3D-images are barely readable.The EDS analysis is clear. However, I expect a comparison with the non-machined area of the surface to prove that the formation of MgO was caused by laser machining.Figure 3 (b) must be improved. The spectrum does not provide sufficient information. Please put a table of atomic percentage of elements instead.The measurement and description of contact angle measurement is clear and does not lack any information. The interpretation of the results is appropriate.The results of the dissolution experiment are clear and precisely described. In figure 5, please add SEM images or 3D micrographs to the corresponding images, if available. Photographs do not allow precise analysis of the morphology other than a rough estimation of coroded area.The discussion of the results is clear and appropriate.

Reply: In the revised manuscript, Figure 2 has been improved. During to the EDS analysis of the atomic percentage of the elements was not accurate, so in this work we didn’t show the table of the atomic percentage.

Conclusion

The major findings of the work are well summarized. However, there is no conclusion or discussion of the results.

How do the results compare to similar approaches, if available? How do the results compare to other methods for the modification of dissolvability of bridge plugs that you mentioned (surface coating, change of material composition, etc.)? Can the presented method be applied to real bridge plugs? Do you expect any changes or improvement in performance when adjusting the DST parameters (e.g. diameter, depth)? 

Reply: In this paper we didn’t compare the DST surface with other surface treatment. The difference between DST and other methods would be studied in the following work. We have applied the presented method to real bridge plugs. We are sure to expect to improve the DST parameters to improvement in performance of the bridge plugs.

Reviewer 2 Report

The paper presents and interesting study in which the connection  to petroleum engineering applications is emphasized. The methods are novel and indeed this study is an interesting cross-disciplinary study. The authors have presented a practical method for controlling the dissolution of an alloy by surface structure modification. However, the presentation of the work in context of petroleum engineering is quite strange. Surely the findings are relevant for other applications in which the dissolution of a Mg-alloy is of interest (i.e. biomaterials). Authors should rewrite their article from a surface engineering perspective and broaden its relvance across multiple disciplines. 

I cannot recommend the publication of this article in a journal that focuses on biomimetics. The link of this system to biomimetics (surface structure being vaguely similar to a lotus leaf) is very weak. 

Author Response

Thanks very much for reviewer's review and kind suggestions.

Reviewer 3 Report

Presentation is clear. I can see the difference and reasoning between samples without DST and with DST. However, the difference with the increasing percentage of DST, especially from 20% to 30%, are not clearly identified and the reasoning is not very clear.  The theory to explain how the air was trapped in the dimple, thus affecting the contact angle, does not explain the difference between 10%, 20% and 30% dimple density.  in Line 118, composition was mentioned as a factor that affects the wettability. Conclusion was drawn in Line 159. Again, more elaboration on this matter can be helpful to improve the overall quality of the paper.  Line 83 and 90: should be 20%, not %20. Some additional check on grammar can be used.  In Figure 4, the blue mark on the upper left corner can be removed to maintain consistency on figures. 

Author Response

Comments: Presentation is clear. I can see the difference and reasoning between samples without DST and with DST. However, the difference with the increasing percentage of DST, especially from 20% to 30%, are not clearly identified and the reasoning is not very clear.  The theory to explain how the air was trapped in the dimple, thus affecting the contact angle, does not explain the difference between 10%, 20% and 30% dimple density.  in Line 118, composition was mentioned as a factor that affects the wettability. Conclusion was drawn in Line 159. Again, more elaboration on this matter can be helpful to improve the overall quality of the paper.  Line 83 and 90: should be 20%, not %20. Some additional check on grammar can be used.  In Figure 4, the blue mark on the upper left corner can be removed to maintain consistency on figures.  

Reply: Thanks very much for your review and good suggestions. In the new manuscript, we have revised some problems as mentioned by the reviewer. The difference with the increasing percentage of DST is not clearly identified and the reasoning is not very clear, we will continue to study these issues in the following work.